# Genome-Wide Identification of *GSTs* Gene Family and Functional Analysis of *BraGSTF2* of Winter Rapeseed (*Brassica rapa* L.) under Cold Stress

**DOI:** 10.3390/genes14091689

**Published:** 2023-08-25

**Authors:** Zaoxia Niu, Lijun Liu, Jinli Yue, Junyan Wu, Wangtian Wang, Yuanyuan Pu, Li Ma, Yan Fang, Wancang Sun

**Affiliations:** 1State Key Laboratory of Aridland Crop Science, Gansu Agricultural University, Lanzhou 730070, China; 15719327746@163.com (Z.N.); wujuny@gsau.edu.cn (J.W.); vampirepyy@126.com (Y.P.);; 2College of Agronomy, Gansu Agricultural University, Lanzhou 730070, China

**Keywords:** *Brassica rapa* L., *GSTs* gene family, *BraGSTF2* gene, over-expression, cold stress

## Abstract

The largest gene families in plants were found to be Glutathione transferases (GSTs), which played significant roles in regulating plant growth, development, and stress response. Within the *GSTs* gene family, members were found to play a crucial role in the low-temperature response process of plants. A comprehensive study identified a total of 70 *BraGSTs* genes. Cluster analysis results demonstrated that the *BraGSTs* in *Brassica rapa* (*B. rapa*) could be categorized into eight sub-families and were unevenly distributed across ten chromosomes. The 39 *BraGSTs* genes were found to be organized into 15 tandem gene clusters, with the promoters containing multiple *cis*-elements associated with low-temperature response. Cold stress was observed to stimulate the expression of 15 genes, with the *BraGSTF2* gene exhibiting the highest level of expression, suggesting its significant involvement in winter *B. rapa*’s response to low-temperature stress. Subcellular localization analysis of the BraGSTF2 protein indicated its potential expression in both the cell membrane and nucleus. The analysis of stress resistance in *BraGSTF2* transgenic *Arabidopsis thaliana* lines demonstrated that the over-expression of this gene resulted in significantly elevated levels of SOD, POD activity, and SP content compared to the wild type following exposure to low temperatures. These levels reached their peak after 24 h of treatment. Conversely, the MDA content was lower in the transgenic plants compared to the wild-type (WT) *Arabidopsis* (*Arabidopsis thaliana* L.). Additionally, the survival rate of *BraGSTF2* transgenic *Arabidopsis* was higher than that of the WT *Arabidopsis thaliana*, suggesting that the *BraGSTF2* gene may play a crucial role in enhancing the cold stress tolerance of winter *B. rapa*. This study lays a foundation for further research on the role of the *BraGSTs* gene in the molecular regulation of cold resistance in winter *B. rapa*.

## 1. Introduction

Winter rapeseed (*Brassica rapa* L.), a type of rapeseed that can withstand winter conditions in northern China, plays a crucial role in ensuring a stable supply of edible oil and improving economic benefits [1,2]. The cold-resistant variety known as Longyou-7 has demonstrated its ability to survive extreme temperatures as low as −30 °C, indicating the presence of numerous cold-resistance genes within it [3,4]. Consequently, Longyou-7 serves as a valuable resource for investigating and understanding the mechanisms underlying cold resistance in winter *Brassica rapa* (*B. rapa*).

Cold stress has been found to have a significant impact on various aspects of plant physiology, biochemistry, metabolism, and molecules, ultimately affecting the growth and development of plants [5,6]. Specifically, the activities of antioxidant enzymes, including superoxide dismutase, glutathione transferase, glutathione reductase, glutathione peroxidase, ascorbate peroxidase, and catalase, have been observed to increase under cold stress. Additionally, the involvement of non-enzymatic antioxidants, such as tripeptide mercaptan and ascorbic acid, has been shown to directly contribute to the stability of cell membranes [7,8,9,10,11,12]. It is worth noting that glutathione transferase (GSTs) is a highly conserved and widely distributed protein family with multifunctional properties, suggesting its ancient origins and broad significance in various biological processes [13].

The GSTs, a multifunctional protein family, have been identified in numerous plant species, with 55 genes found in *Arabidopsis* [14], 179 in *Brassica napus* [15], 84 in *Hordeum vulgare* [16], 59 in *Gossypium raimondii* [17], 75 in *Brassica rapa* [18], 65 in *Brassica oleracea* [19], and 23 in *Citrus sinensis* [20]. These *GSTs* genes play a crucial role in plant secondary metabolism, growth, and development, particularly in response to biotic and abiotic stress [21,22]. For instance, the *AtGSTU17* gene in *Arabidopsis* regulates seed development, including hypocotyl elongation and anthocyanin accumulation [16]. Additionally, the transfer of the *VvGSTF13* gene enhances Grape’s salt and drought tolerance [23]. The *ThGSTZ1* gene has been found to have a dual role in enhancing tolerance to drought and salt stress, as well as improving antioxidant activity through the regulation of reactive oxygen species (ROS) metabolism in plants [24,25]. *GSTs* genes have been extensively studied for their involvement in gene expression related to salt stress tolerance [26].

While there is substantial research on the impact of *GSTs* genes on oxidative and temperature stress tolerance, limited research has been conducted on their role in cold resistance in winter *B. rapa*. This study aims to comprehensively analyze the expression and function of *GSTs* genes, including *GSTF2*, under low-temperature stress conditions. The purpose of this study was to conduct a comprehensive analysis of the number of *GSTs* gene families and the function of the *BraGSTF2* gene under low-temperature stress in winter *B. rapa* at the whole genome level. This research aimed to establish a theoretical foundation for analyzing the regulatory mechanism of the gene family under cold stress and abiotic stress in winter *B. rapa*.

## 2. Results

### 2.1. Identification and Protein Characterization of BraGSTs Gene Family

The present study focused on the identification and characterization of the *BraGSTs* gene family in *B. rapa*. A comprehensive search was conducted using specific keywords and the Pfam number in the *B. rapa* database. These proteins were identified as having the reported peroxidase domains after sequence analysis using SMART, CDD, and Pfam. As a result, a total of 70 members belonging to the *BraGSTs* gene family were successfully identified. Additionally, the members of the *BraGSTs* gene family were named based on their respective chromosomal positions. Furthermore, an analysis of the physical and chemical properties of all BraGSTs proteins was conducted. This analysis revealed that the number of amino acids in these proteins ranged from 114aa to 600aa. Moreover, the theoretical isoelectric point (PI) of BraGSTs proteins varied from 4.84 to 9.54. The stability index exhibited a range of values spanning from 27.07 to 95.27. Similarly, the fat index displayed a range from 70.12 to 106.00. The hydrophilic and hydrophobic range encompassed values ranging from −0.52 to 0.16. The primary constituents of the BraGSTs protein were identified as carbon (C), hydrogen (H), nitrogen (N), oxygen (O), and sulfur (S). The analysis of the protein’s secondary structure prediction revealed that the BraGSTs proteins predominantly consisted of α helices, β sheets, random coils, and extended chains (Appendix A).

### 2.2. Phylogenetic Analysis, Chromosome Distribution, and Gene Duplication in BraGSTs Gene Family

The NJ method was used to construct a cluster map that divided 70 *GSTs* genes into 8 main subfamilies in *B. rapa* (Figure 1). These subfamilies include the γ-subunit of translation elongation factor (EF1G), Phi (GSTF), Tau (GUTF), Zeta (GSTZ), DHAR, Lambda (GSTL), Theta (GSTT), and TCHQD. Among these subfamilies, the 18 *BraGSTs* genes were classified as Phi (GSTF), the 39 *BraGSTs* genes as Tau (GSTU), the four *BraGSTs* genes as DHAR, and one *BraGSTs* gene as TCHQD. Moreover, the Zeta, Lambda, Theta, and EFIG subfamilies each consisted of two distinct *BraGSTs* genes (Table 1).

Based on the distribution map of the *BraGSTs* gene family on chromosomes in *B. rapa*, as depicted in Figure 2, it is evident that the genes were dispersed across ten distinct chromosomes. Notably, Chromosome 7 exhibits the highest number of gene distributions, with 14 *BraGSTs* genes, while Chromosome 1 harbors only one *BraGSTs* gene. In comparison to other subfamilies, the Tau family members display a wide distribution pattern, spanning nine chromosomes (A01, A02, A03, A04, A05, A06, A07, A08, A09).

Tandem repeats play an important role in expanding the gene family during the evolutionary process. In this study, a total of 70 *BraGSTs* genes were subjected to sequence alignment in order to identify genes with tandem repeats. The analysis revealed that 38 genes (55.7%) formed 15 tandem repeat clusters. Among these clusters, the A09 chromosomes exhibited the highest number of tandem repeat clusters, with a total of four clusters. Additionally, the A02, A03, A06, and A07 chromosomes each contained two tandem repeat clusters. On the other hand, the A04, A05, and A10 chromosomes harbored only one tandem repeat cluster each, while no tandem repeats were observed on the A01 and A08 chromosomes. This implies that tandem repeats have a significant impact on the proliferation of the *BraGSTs* gene family.

### 2.3. Conservative Structure and Gene Structure Analysis of BraGSTs Gene Family

The conserved motifs in *B. rapa* were analyzed using the Meme online tool (MEME—Submission form (meme-suite.org) accessed on 12 January 2023) (Figure 3B). The analysis predicted a maximum of 10 conserved motifs, with all other settings set to default values. A total of 10 conserved motifs were identified in *B. rapa*. The *BraGSTs* gene family was also identified, and the amino acid length of each motif ranged from 15 to 64 amino acids (Table 2). Among these motifs, Motif 5 was found to be highly conserved and present in almost all members of the *BraGSTs* gene family. This result suggests that Motif 5 plays an essential role and has important functions in the *BraGSTs* gene family.

The gene structure maps of *BraGSTs* were generated using TBtools software(TBtools v1.132). *BraGSTU8* lacked a UTR, while *BraGSTF11* contained one UTR. All members of the *BraGSTs* gene family possessed a coding sequence (CDS). The quantity of CDS varied significantly among different *BraGSTs* genes. Specifically, *BraGSTU19* had one CDS, *BraGSTL1* within the Lambda subfamily had 15 CDS, and *BraGSTL2* had 8 CDS, making it the subfamily with the highest intron distribution. These results indicate that the DHAR and Phi subfamilies in *B. rapa* exhibit greater conservation.

### 2.4. Analysis of cis-Acting Elements in BraGSTs Gene Family

A total of 63 cis-acting elements were identified within the promoter region of the *BraGSTs* gene family. These elements were categorized into nine functional groups, with light response elements comprising the majority, followed by promoter and enhancer elements, low-temperature stress response elements, and plant hormone response elements (Figure 4). Notably, *CAAT-box* and *TATA-box* were identified as the most significant elements within promoters and enhancers, while LTR and ARE constituted the largest proportion of cryogenic response components. The ABRE and CGTCA-motif response elements were the most prevalent in plant hormone responses. The presence of MYB, MYC, and MYB-like response elements was abundant in both biotic and abiotic stress responses. Certain genes exhibit endosperm-specific expression elements, namely GCN4-motif and AACA-motif. Additionally, there were genes with meristem-specific expression elements, such as CAT-box and RY-element. Some genes possess protein binding sites, including AT-rich elements and BOX III. The response elements associated with low-temperature stress encompass ARE, GC-motif, GTGGC-motif, LTR, MBS, TC-rich, W box, and WUN-motif.

### 2.5. Expression of BraGSTs Gene Family in B. rapa under Low-Temperature Stress

During the early stage of laboratory experimentation, RNA-seq analysis revealed that the majority of *BraGSTs* genes exhibited no response. However, a subset of 15 *BraGSTs* genes demonstrated involvement in low-temperature stress (|log2FC| > 1). Among these, six genes (*BraGSTF6*, *BraGSTF7*, *BraGSTF14*, *BraGSTU9*, *BraGSTU12*, and *BraGSTU15*) exhibited significant up-regulation at both the S1 and S4 stages. Additionally, *BraGSTF2*, *BraGSTF3*, *BraGSTF18*, *BraGSTU14*, *BraDHAR2*, and *BraDHAR3* were significantly up-regulated at S4. Conversely, *BraGSTF10* and *BraDHAR4* were significantly down-regulated at S4, while *BraGSTU10* demonstrated significant up-regulation at S1 (see Figure 5).

### 2.6. Subcellular Localization of BraGSTF2 Protein

The tissue and cell sites where the BraGSTF2 protein was predicted using an online website (https://wolfport.hgc.jp/, accessed on 12 January 2023). The prediction results indicate that the BraGSTF2 protein is localized in the cytoplasm and cell membrane. To further investigate its subcellular localization, the 35S-BraGSTF2-GFP subcellular localization vector was introduced into tobacco plants, while the 35S-GFP empty vector served as the control. The plants were incubated overnight for documentation purposes. After 2–3 days, the epidermis was dissected, and the fluorescence was visualized using a laser confocal microscope. The obtained results are presented in Figure 6. Furthermore, alongside the conspicuous green fluorescence observed in the cell membrane, the BraGSTF2 protein exhibits evident fluorescence within the nucleus, while no fluorescence is detected in other cellular compartments. These findings indicate that the expression of the BraGSTF2 protein in tobacco leaves is likely localized within the cell membrane and nucleus, aligning with the anticipated outcomes. However, additional validation of alternative techniques is warranted in subsequent investigations.

### 2.7. Screening and Identification of Transgenic Arabidopsis in BraGSTF2 Gene

Following the infection of *Arabidopsis*, the plants were subjected to antibiotic treatment for three successive generations. The T3 generations of transgenic *Arabidopsis* plants were subsequently transplanted into flowerpots, where they exhibited normal growth for the subsequent experiment (Appendix A). Positive plants with overexpression of *BraGSTF2* yielded target bands of approximately 700 bp, whereas wild-type *Arabidopsis* plants and water samples did not exhibit band amplification. This observation confirmed the successful transfer of the *BraGSTF2* target gene into *Arabidopsis*, resulting in the generation of a positive transgenic plant (Appendix A).

The RNA of four transgenic single plants overexpressing *BraGSTF2* was extracted and quantitatively analyzed using fluorescence. The results demonstrated that the relative expression of the *BraGSTF2* gene in the overexpressed plants was significantly higher compared to the WT. Notably, the G1 single plant exhibited the highest relative expression, which was 22.9 times that of the wild type. Conversely, the G3 single plant displayed the lowest relative gene expression, 6.8 times that of the wild type. The relative gene expression of *GSTF2* in the G3 single plant was found to be the lowest, measuring 6.8 times that of the WT. This observation, along with the fact that the other three single plants exhibited a relative gene expression of *GSTF2* 22.9 times that of the WT, suggests that all four single plants can be classified as *BraGSTF2* overexpressed plants (Appendix A).

### 2.8. Phenotypic and Survival Analysis of WT and BraGSTF2 Transgenic Arabidopsis under Low-Temperature Stress

WT and transgenic plants were subjected to a −4 °C treatment for varying durations of 3 h, 6 h, 12 h, and 24 h, followed by normal growth conditions for 7 days. The results demonstrate that both WT and transgenic plants exhibited normal growth after being exposed to low temperatures for 3 h and 6 h. However, the WT plants subjected to a 12 h treatment experienced significant mortality, whereas the majority of the transgenic plants managed to survive under the same low-temperature conditions. Furthermore, a small proportion of transgenic plants survived even after 24 h exposure to low temperatures, as illustrated in Figure 7. The results of the statistical analysis conducted on the survival rate of seedlings subjected to low-temperature stress indicate that both WT and transgenic plants treated at −4 °C for 3 h and 6 h exhibited normal growth. However, a significant disparity in survival rates was observed between WT and transgenic plants after 12 h of treatment. Specifically, the survival rate of WT plants was recorded at 2.1%, whereas the survival rate of transgenic plant seedlings reached 75.1% (Figure 8). Furthermore, all WT plants perished after 24 h, while the survival rate of transgenic plants was measured at 43.3%.

### 2.9. Physiological and Biochemical Index Analysis of Arabidopsis under Low-Temperature Stress

In this study, *Arabidopsis* leaves were utilized to assess physiological and biochemical indicators to evaluate the impact of low-temperature treatment at −4 °C on plant damage. The findings revealed that the activities of CAT and SOD, as well as the SP content, initially increased and subsequently decreased in both wild-type (WT) and transgenic plants. Conversely, the MDA content consistently increased throughout the low-temperature stress. At 25 °C, the CAT activity, SOD activity, and MDA content of the WT plants exhibited minimal changes in comparison to the transgenic plants. However, the SP content of the WT plants was higher than that of the transgenic plants.

After a 3 h low-temperature treatment, the catalase (CAT) activity, superoxide dismutase (SOD) activity, and soluble protein (SP) content of the wild type (WT) were found to be higher than those of the transgenic plant (G), exhibiting a 52.6% increase compared to the initial measurement. Conversely, the malondialdehyde (MDA) content of the WT was significantly higher, with a 51.5% difference compared to the transgenic plant (G). However, the CAT activity, SOD activity, and SP content did not show a significant increase after 6 h of low-temperature treatment compared to the initial measurement. Following a 6-h low-temperature treatment, the WT exhibited a 12.2% increase in CAT activity, a 16.7% increase in SOD activity, and an increase in SP content compared to the transgenic plant (G).

Following a 12 h treatment at −4 °C, the activities of CAT, POD, and the content of SP reached their peak values, exhibiting a respective increase of 16.1%, 32.8%, and 11.4% compared to the WT. However, as the processing time prolonged, the activities of CAT, SOD, and the content of SP declined, while the content of MDA continued to rise (Figure 9).

## 3. Discussion

Glutathione S-transferase plays an important role in plant growth and development and stress management [27]. Genome-wide analysis indicated that 70 *GSTs* genes from *B. rapa* were divided into eight subclasses based on the domain information and phylogenetic analysis. The Tau subclass was numerous, with 37 genes and Phi with 22 (Figure 1 and Table 1). All genes are unevenly distributed on ten chromosomes (Figure 2). Most genes in the same evolutionary branch contained the same conserved motif, which was similar to that of *B. rapa* (Figure 3B).

The subclass of exon and intron in the *GSTs* gene family was determined through cluster analysis and gene structure analysis (Figure 3C). Previous studies have demonstrated the significant involvement of these two types of *GSTs* genes in plant resistance to abiotic stress [28]. Furthermore, a hormone stress-specific cis-regulatory element was identified in the promoter region of the *GSTs* gene, suggesting its involvement in hormone regulation, as illustrated in Figure 4. These findings establish a theoretical foundation for future investigations into the functionality of the *GSTs* gene family, particularly in eukaryotes.

To comprehend the reaction of *GSTs* genes to low-temperature stress, the expression levels of 15 genes were observed to be significantly altered, either upregulated or downregulated (Figure 5). Cold-tolerant varieties exhibited elevated GST enzyme activity and expression levels of 28 stress-related genes when subjected to low-temperature stress. Certain *CmGSTs*, specifically those classified as Tau, Phi, and DHAR, were found to play a crucial role in response to cold stress. The study suggests that the *JrGSTU1* transgenic walnut has the potential to enhance its cold tolerance by upregulating the expression of the *GSTs* gene and increasing GST enzyme activity under cold stress conditions [29]. Additionally, Ma et al. [4] observed a significant upregulation of BnaA02g35760D, BnaC06g20450D, BnaC06g35490D, BnaA02g03230D, and BnaA02g35980D genes in strong cold-resistant walnut varieties, with expression levels 7–12 times higher compared to weak cold-resistant varieties under cold stress. The study found that *Arabidopsis* plants overexpressing *GSTF2* under low-temperature stress were stronger than WT plants, and overexpressing *GSTF2* in Korean rape enhanced the tolerance of plants [30]. Harshavardhanan et al. found that under low-temperature stress, the expression of *BoGSTF10* and *BoGSTU24* genes in cabbage was up-regulated [19]. In the third chapter, the proteomic study found that the protein expression abundance of the *GSTF2* gene in Longyou-7 was 1.6. The *GSTF2* gene also participated in the response to low-temperature stress, but its functional mechanism needs further study.

The subcellular localization findings indicated that *GSTF2* exhibited presence in both the cell membrane and the nucleus, as depicted in Figure 6. Consequently, further investigation is required to elucidate its specific functionalities. To this end, the *GSTF2* gene was introduced into *Arabidopsis*, resulting in overexpression. Subsequent exposure to low-temperature stress revealed that the transgenic plants exhibited a higher survival rate in comparison to the wild type, as illustrated in Figure 7 and Figure 8. Subsequent analysis of physiological and biochemical parameters in the transgenic plants subjected to low-temperature stress demonstrated a significant increase in enzyme activity and soluble protein content relative to the overexpressed plants (Figure 9). Previous research conducted on *Arabidopsis* has demonstrated that *AtGSTF2* lacks enzymatic catalytic activity but functions as a ligand-protein by directly binding with auxin [31]. This interaction plays a crucial role in regulating the distribution and accumulation of auxin within plants. Furthermore, this finding provides evidence for the association between auxin and the differentiation of stem tip meristem [32]. Additionally, it was observed that these plants exhibit remarkable tolerance to freezing conditions, enabling them to adapt effectively to such harsh environmental circumstances.

The findings of this study demonstrate that the expression of the *GSTF2* gene in transgenic plants was significantly elevated compared to the wild type under low-temperature stress [33]. This study further supports the notion that the *GSTF2* gene plays a crucial role in enhancing the cold resistance of plants. Consequently, these results offer valuable insights into the underlying mechanisms of cold resistance in winter *B. rapa* and provide a foundation for enhancing the cold resistance of northern winter rapeseed.

## 4. Materials and Methods

### 4.1. Identification of Members of BraGSTs Gene Family in B. rapa

The *B. rapa* genome and gene structure annotation information, exhibiting significant homology, were acquired from the *B. rapa* database (http://brassicadb.cn/, accessed on 12 January 2023) [34]. The BraGSTs domain (PF00043 and PF02798) was obtained from the Pfam database (https://www.ebi.ac.uk/interpro/, accessed on 12 January 2023) [24]. The HMMER software was utilized to search the Longyou-7 database, resulting in the retrieval of protein sequences containing the PF00043 and PF02798 domains [34]. At the same time, the online software ExPASy (http://web.expasy.org/protparam/, accessed on 12 January 2023) was used to predict the amino acid number, molecular weight, isoelectric point, hydrophilicity, and fat index of all family members [35,36]. ClustaW was used for multi-sequence alignment of BraGSTs amino acid sequences. Based on multiple sequence alignment, the cluster diagram is constructed by the adjacency method (Neighbor-Join, NJ) in MEGA7.0 software [37].

The initial coordinates of each *BraGSTs* gene on the chromosome, along with other pertinent details, were acquired from the *B. rapa* genome database. The physical positioning of these genes on the chromosome was then mapped using MapChart software [38]. The *BraGSTs* gene sequences were obtained, and the promoter sequences of candidate *BraGSTs* genes in *B. rapa*, spanning 2000 bp upstream, were extracted. The cis-acting elements were subsequently analyzed using the PlantCARE database (http://bioinformatics.psb.ugent.be/webtools/plantcare/html/, accessed on 12 January 2023), and the corresponding charts were generated [39]. The conserved motif of the BraGSTs protein in *B. rapa* was examined using the online analysis tool MEME (http://meme-suite.org, accessed on 12 January 2023). A maximum of 10 motifs were considered, while the default parameters were utilized for all other settings [40]. Fragment replication and tandem replication of the *B. rapa GSTs* gene family were analyzed using MCScanX software (made with default parameter values) [41].

### 4.2. Prediction of BraGSTs Expression under Low-Temperature Stress

The expression of the *BraGSTs* gene family was predicted and analyzed using the transcriptome data of “Longyou-7” provided by the rape laboratory of Gansu Agricultural University. Additionally, the heat map was generated using TBtools software [42]. The time points S1 and S4 corresponded to the overwintering period on 13 October (0 °C) and 16 December (−11 °C), respectively.

### 4.3. Preliminary Verification of BraGSTF2 Gene Function

The *BraGSTF2* gene was cloned by PCR based on specific primers (*BraGSTF2*-F: ATGGCAGGTATCAAAGTTTTCG; *BraGSTF2*-R: CTGAAGGATCTTCTGTGAAGC). Concerning Homologous Recombination technology, PCR amplification was carried out using primers (*BraGSTF2*-F: CGGGGGACGAGCTCGGTACCATGGCAGGTATCAAA GTTTTCG; *BraGSTF2*-R: ACCATGGTGTCGACTCTAGACTGAAGGAT CTTCTGTGAA GC). The recovered products were ligated with BP Clonase enzyme (Invitrogen, Carlsbad, CA, USA) and the pDONR vector. After transforming into coliform bacteria (DH5α), the positive clones were screened by smear sequencing. The positive clones were selected to interact with LR Clonase enzyme (Invitrogen, Carlsbad, CA, USA) to connect to the overexpression vector pEarly-Gate101, following the instructions for Agrobacterium tumefaciens GV3101 (Anyu Biotechnology Co., Ltd., Shanghai, China). After the bacteria were picked out from the coated plate, colony PCR was carried out, and the positive bacterial solution was selected and preserved [43].

Transgenic *Arabidopsis* was obtained by the floral-dip method. The T3 seeds were obtained by screening with antibiotics. The T3 seeds of transgenic *Arabidopsis* lines and WT *Arabidopsis* were sown in pots according to the experimental requirements. The phenotypes of WT and transgenic *Arabidopsis* plants treated at −4 °C were observed at the seedling stage. WT and transgenic plants were grown for about 30 days and were treated in a −4 °C incubator for 3 h, 6 h, 12 h, and 24 h, respectively. Four plants were taken in each pot, and the other part resumed normal growth, and the plant phenotype and survival rate were observed and counted after one week. The leaves of the plants were frozen in liquid nitrogen and stored in the cryogenic refrigerator. The CAT activity and SOD activity, MDA content, and SP content were determined, and the rest of *Arabidopsis* recovered. One week later, the plant survival rate was observed and counted [44,45]. The CAT activity and SOD activity, MDA content, and SP content were estimated according to the method developed by Gill et al. [46] and Quiroga et al. [47]. Three biological replicates were set up for each of the above experimental treatments.

### 4.4. RNA Isolation, Reverse Transcription, qRT-PCR, and T Transcriptome Expression Analysis

RNA Extraction Kits (TIANGEN Biotech Co., Ltd., DP432, Beijing, China) were employed to extract total RNA from both wild-type (WT) and transgenic plants, ensuring the elimination of genomic DNA contamination. The cDNA synthesis was carried out using the PrimeScript™ RT Master Mix (Vazyme, Nanjing, China) following the provided instructions. Subsequently, the quality and concentration of the obtained cDNA were assessed using an ultra-trace UV-visible spectrophotometer (NanoVueTM Plus, Wilmington, DE, USA) and stored in a standby state.

Genes within the transcriptome data were chosen, and specific primers were designed for qRT-PCR based on the guidelines provided by the ChamQ Universal SYBR qPCR Master Mix (Shanghai, China). Primers (*BraGSTF2*-pcr-F: GCGAACTGTCTGATGCTC and *BraGSTF2*-pcr-R: CGTATCGGTGAGCTATGTACTG) were used to perform qRT-PCR on *Arabidopsis*. AtActin2 (F: TGTGCCAATCTACGAGGGTTT; R: TTTCCCGCTCTGCTGTTGT) was used as an internal control for Arabidopsis. The reaction procedure consisted of an initial denaturation step at 95 °C for 10 min, followed by 40 cycles of denaturation at 95 °C for 30 s and annealing/extension at 60 °C for 1 min. Subsequently, a melting curve analysis was conducted within the temperature range of 65–95 °C. The qRT-PCR efficiency of the genes was determined by evaluating the standard curve derived from the gradient dilution of cDNA. To normalize the quantity of template cDNA, the gene fragment encoding *Arabidopsis* actin RNA was utilized as an internal control. The relative expression values of each gene were calculated using the comparative 2^−∆∆CT^ method [48]. The RNA-seq libraries (SRP179662) and (SRP211768) were selected for gene-expression analysis [44,45]. Three biological replicates were set up for each experimental treatment.

### 4.5. Subcellular Localization of BraGSTF2 Protein in Tobacco

Agrobacterium tumefaciens strains containing the PCAMBIA2300-BraGSTF2-GFP construct, and the pCAM-BIA2300-GFP empty vector were transiently introduced into Benji tobacco leaves. The tobacco plants were cultured at 23 °C under dark conditions for 24 h. Subsequently, small sections of tobacco leaves were excised to create thin slices for analysis. The fluorescence patterns of the pCAMBIA2300-BraGSTF2-GFP and pCAM-BIA2300-GFP constructs were visualized using a laser scanning confocal microscope LSCM800 (Carl Zeiss, Jena, Germany) with an excitation wavelength of 488 nm [49].

## 5. Conclusions

In this study, a total of 70 members of the *BraGSTs* gene family were identified and classified into 8 subfamilies. It was observed that members within the same subfamily exhibited identical gene structures and functional motifs. Furthermore, the distribution of these genes across 10 chromosomes was found to be uneven. Based on these findings, it is hypothesized that the *BraGSTF2* gene plays a significant role in mitigating the effects of low-temperature stress. Additionally, the subcellular localization of the BraGSTF2 protein was determined to be in the nucleus and cell membrane. Further investigation revealed that the over-*Arabidopsis BraGSTF2* gene resulted in higher levels of CAT, SOD activity, and SP content compared to the wild type, while the MDA content was lower under low-temperature stress conditions. These results suggest that the *BraGSTF2* gene may have a regulatory function in response to low-temperature stress.

## Figures and Tables

**Figure 1 genes-14-01689-f001:**
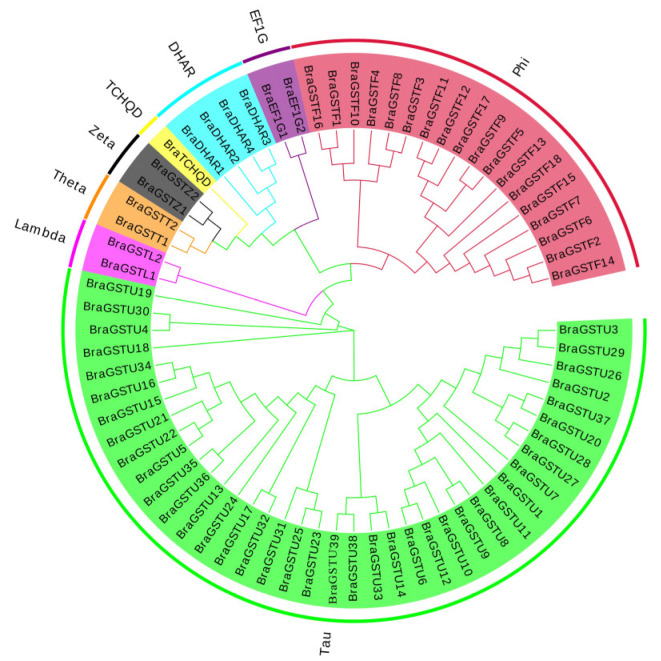
Clustering analysis of *BraGSTs* gene family in *B. rapa*. Eight subfamilies and branches are marked with different colors.

**Figure 2 genes-14-01689-f002:**
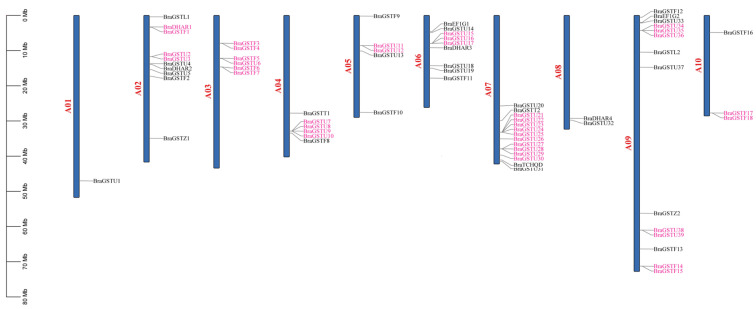
Chromosomal location and gene duplication in *B. rapa*. The tandem duplicated genes are marked with hot pink. The chromosome number is indicated at the top of each chromosome; Chr00 is the gene scaffold. The scales on the left and right sides (in Mb) are for the chromosomes and scaffold, respectively.

**Figure 3 genes-14-01689-f003:**
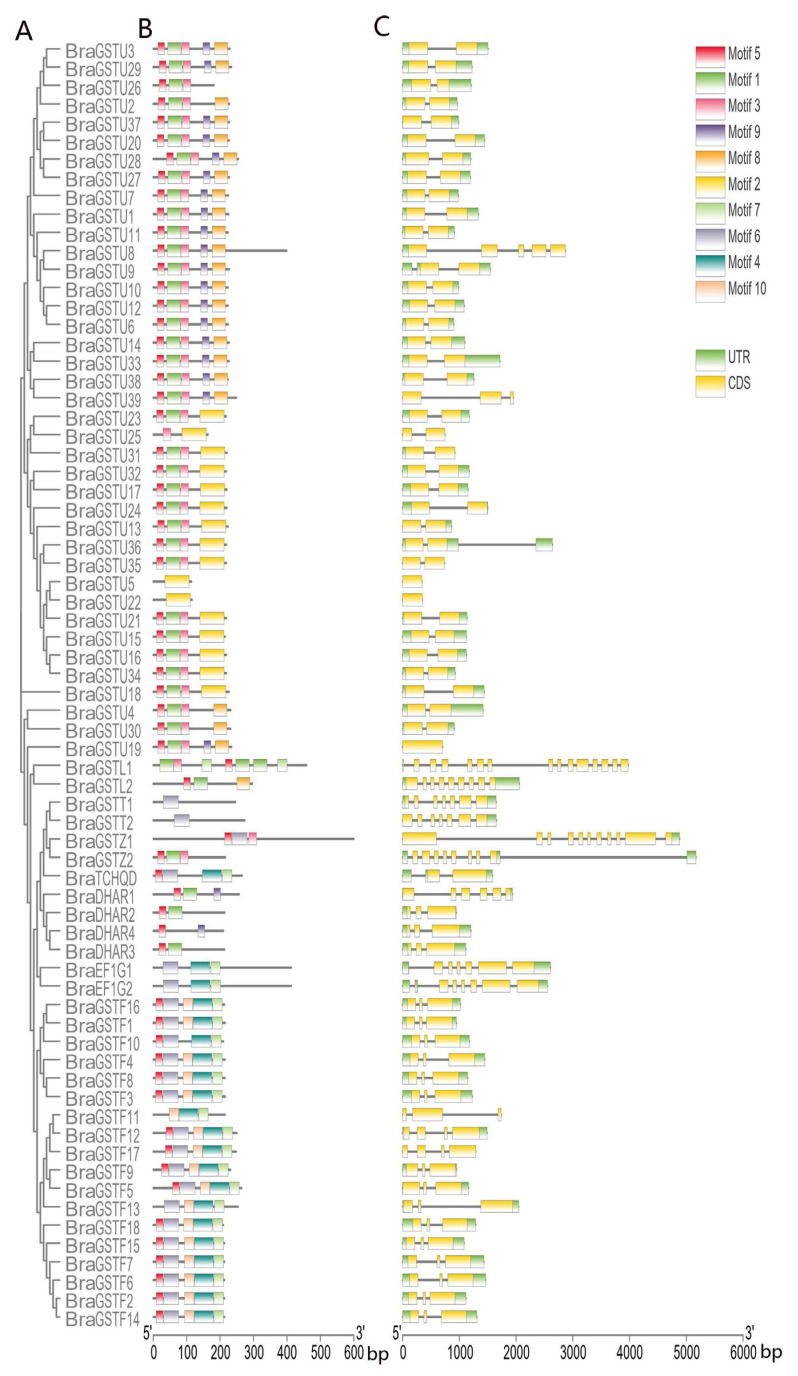
Clustering analysis, conserved structure, and gene structure analysis of the *BraGSTs* gene in *B. rapa*. (**A**) Using MEGA7 to draw a clustering analysis using the maximum likelihood method with default parameters in BraGSTs proteins. (**B**) Distribution of conserved motifs in the BraGSTs proteins. Boxes of different colors represent 10 putative motifs. (**C**) Exon/intron organization of the *BraGSTs* genes. The yellow box indicates the CDS, the green box indicates the UTR, and the black line indicates the intron—scale bars represented as bp.

**Figure 4 genes-14-01689-f004:**
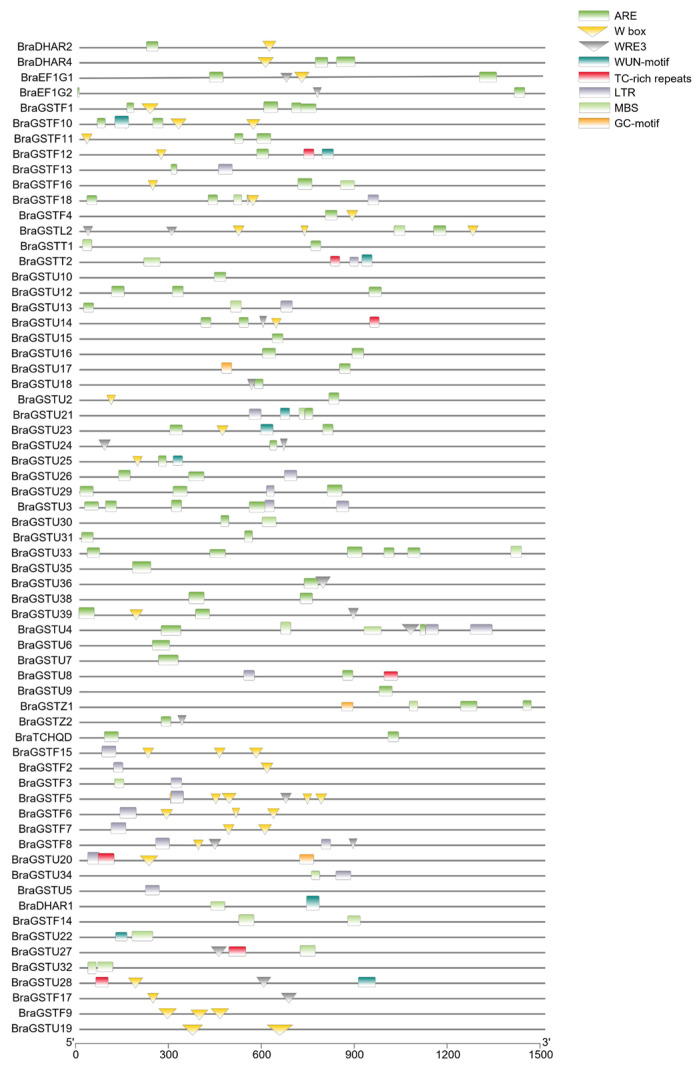
Frequency of cis-regulatory elements in the upstream regions of BraGSTs protein in *B. rapa*. Scale bars represent bp.

**Figure 5 genes-14-01689-f005:**
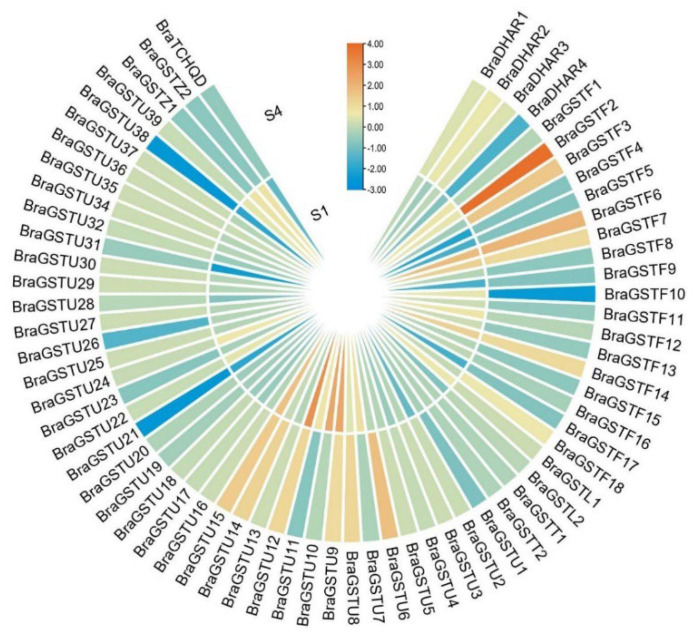
The heat map of *BraGSTs* genes expression profile under freezing stress in *B. rapa*. The values of RNA-seq were transformed using log2, and the heat map was generated by TBtools software. The S1 and S4 represented the overwintering period on 13 October (0 °C) and 16 December (−11 °C). The color scale represents the relative expression levels from low (blue) to high (red).

**Figure 6 genes-14-01689-f006:**
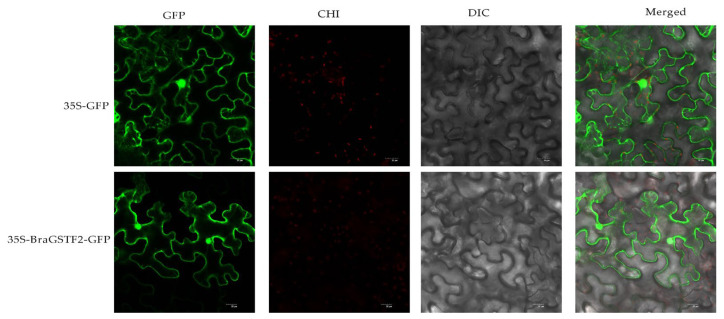
The figures presented above portray the subcellular localization of the BraGSTF2 protein within tobacco cells. Progressing from left to right, these figures illustrate the observations of subcellular localization for the 35S-GFP empty vector in distinct fields: dark field, bright field, and the merged field, respectively. Similarly, the figures below showcase the subcellular localization observations of the 35S-BraGSTF2-GFP recombinant vector in three distinct fields, also arranged from left to right. For reference, the scale bars indicate a length of 20 μm, facilitating size assessment. In this context, GFP corresponds to the green fluorescent protein, DIC pertains to bright field microscopy, and Merged signifies an overlaid photograph combining two channels to provide a comprehensive visual representation.

**Figure 7 genes-14-01689-f007:**
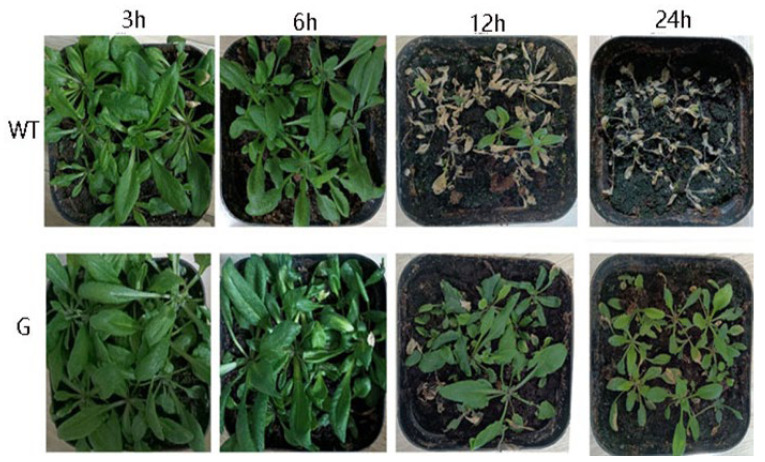
The phenotype of WT and *BraGSTF2* transgenic *Arabidopsis* after low-temperature stress. WT: wild type, G: *BraGSTF2* transgenic *Arabidopsis*.

**Figure 8 genes-14-01689-f008:**
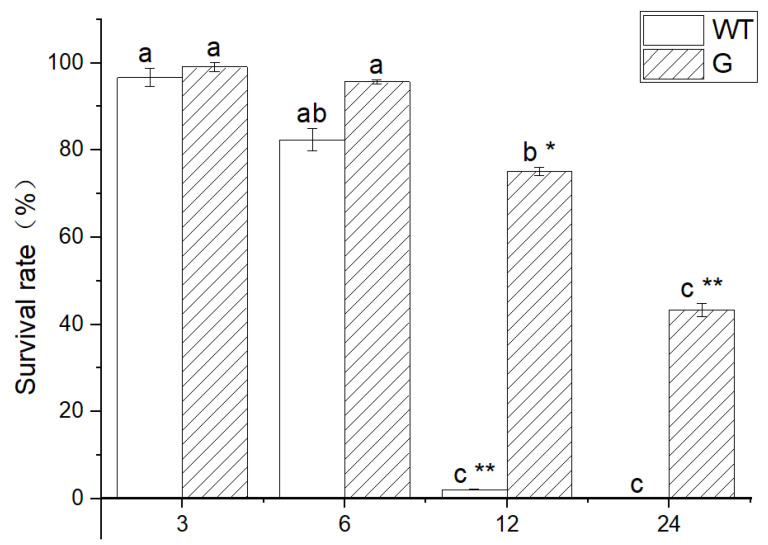
The survival rate of plants after low-temperature stress. WT: wild type, G: *BraGSTF2* transgenic *Arabidopsis*; Different lowercase letters represent significant differences in low-temperature treatment times. * indicated that there were significant differences among varieties (*p* < 0.05), and ** indicated that there were extremely significant differences among varieties (*p* < 0.01). The error bar represents the standard error of the average of the sample.

**Figure 9 genes-14-01689-f009:**
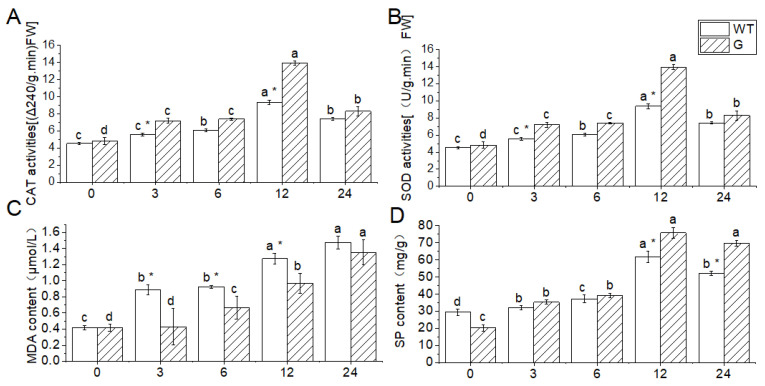
Effect of low-temperature stress on physiological indexes in transgenic and WT plants. (**A**) CAT activities; (**B**) SOD activities; (**C**) MDA content; (**D**) SP content. WT: wild type, G: *BraGSTF2* transgenic *Arabidopsis*; Different lowercase letters represent significant differences in low-temperature treatment times. * indicated that there were significant differences among varieties (*p* < 0.05). The error bar represents the standard error of the average of the sample.

**Table 1 genes-14-01689-t001:** The *BraGSTs* gene family was classified based on their domain and phylogenetics in *B. rapa*.

Subfamilies	Number of Identified Genes
Phi	18
Tau	39
Zeta	2
Theta	2
Lambda	2
DHAR	4
EF1G	2
TCHQD	1
Total	70

**Table 2 genes-14-01689-t002:** Amino acid sequence and amino acid number of Motif.

Motif	Amino Acid Sequence	Number of Amino Acids
Motif 1	EEDLGNKSELLLESNPVHKKIPVLIHNGKPICESLIIVEYIDETWP	46
Motif 2	FGYVDIALIGFYSWFDAYEKFGNFSIEAECPKLIAWAKRCLKRESVAKSLPDSEKVVEYVPELR	64
Motif 3	GNPJLPSDPYERAQARFWADFIDEKV	26
Motif 4	FKPVYGLTTDQAVVKEEEAKLAKVLDVYEARLKESKYLAG DTFTLADLHHJPVIQYLL	58
Motif 5	SPFSRRVRJALELKGVPYE	19
Motif 6	KZFDELLKTLESELGDKPYFGGET	24
Motif 7	AKKEFIELLKTLEKELGDKTYFGGET	26
Motif 8	EEVKLLGYWPSPFSM	15
Motif 9	TPTKKLFEERPHVNEWVAEITARPAW	26
Motif 10	CLALLEEAFQKSSKGKGFFGGENIGFLDIACGSFLG	36

## Data Availability

Data is contained within the article and Appendix A.

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
