# Peer review of "Genome-Wide Identification of GSTs Gene Family and Functional Analysis of BraGSTF2 of Winter Rapeseed (Brassica rapa L.) under Cold Stress"

_genes, 2023, doi:10.3390/genes14091689_

Round 1

Reviewer 1 Report (Previous Reviewer 2)

The manuscript entitled "Genome-wide Identification of GSTs Gene Family and Functional Analysis of BraGSTF2 of Winter Rapeseed 3 (Brassica rapa L.) under Cold Stress" depicts the overall organisation of the Get-encoding gene family and further dissects the involvement of the BraGSTF2 in the response of plants to cold stress.

I am pleased to read the revised manuscript with a huge improvement of the english, making it understandable. However, I still propose that a thorough English and grammar revision is needed.

There are many aspects to correct, like in line(s):

- 15 - BraGST, when referring to gene(s), should be written in italic (here and throughout the text);

- 16 - Species scientific name should always be written in italic. When referring to the same species for the following times, an abbreviation in italic should be used instead (Brassica rapa -> B. rapa);

-18 - "cis" should be in italic (and throughout the text);

- 23 - put in italic Arabidospis thaliana here and throughout the text;

- there should be no paragraphs in the abstract.

- 35 - it is L.;

- 69 - why is BraGSTF2 underlined?

- 105 - explain how 2 genes can be divided into 4 sub-families… Was it a typo?

- 115 - include in the caption what the red boxes are.

- 136-137 - indicate units for bars/scales

- 161 - where is the pie chart? Also, include units for scales

- 175-178 - merge text with caption

- 196 - introduce bar/scale for cell dimensions

- 197-201 - merge text with caption

- 202 - screening, not svreening

- 203 - infection, not infestation

- 213-214 - How come WT Arabidopsis also express the transgene BraGSTF2? This result makes me uneasy and leads me to not trust these results. Authors should sequence the qPCR products of the transgenic and WT lines to clarify why WT plants also have an amplicon and if it corresponds to the same gene.

- 244-249 - merge text with caption

- 387 - Four, instead of 4

- 393-394 - provide references for enzyme activities and the other biochemical determinations, providing that you did not make alterations to those protocols. If alterations were done, please describe them.

- 398-404 - format text accordingly

- 405-406 - were these primes those described in lines 389/390? Clarify in text.

- 441 - genes, not gene.

- References - revise species' scientific names and put them in italic when needed. Also do the same for genes' names.

I still propose that a thorough English and grammar revision is needed.

Author Response

Response to Reviewer 1 Comments

Point 1- 15 - BraGST, when referring to gene(s), should be written in italic (here and throughout the text);- 16 - Species scientific name should always be written in italic. When referring to the same species for the following times, an abbreviation in italic should be used instead (Brassica rapa -> B. rapa);-18 - "cis" should be in italic (and throughout the text);- 23 - put in italic Arabidospis thaliana here and throughout the text;- there should be no paragraphs in the abstract.- 35 - it is L.;

Response 1: Thank you for your valuable suggestions. We have revised and corrected fixed all these problems.

Point 2: - 69 - why is BraGSTF2 underlined?

Response 2: Thank you for providing suggestion. We have fixed all this problem.

Point 3: - 105 - explain how 2 genes can be divided into 4 sub-families… Was it a typo?

Response 3: Thank you for your comment. Two BraGSTs genes are divided into Zeta, Lambda, Theta, and EFIG subfamilies.

Point 4: - 115 - include in the caption what the red boxes are.

Response 4: Thank you for your comment. We have fixed this problem. The tandem duplicated genes are marked with red boxes. We have fixed all this problem.

Point 5: - 136-137 - indicate units for bars/scales

Response 5: Thank you for your suggestion. We have fixed all this problem.

Point 6: - 161 - where is the pie chart? Also, include units for scales

Response 6: Thank you for your comment. We have added the pie chart and include units for scales.

Point 7: - 175-178 - merge text with caption

- 197-201 - merge text with caption;

- 244-249 - merge text with caption

Response 7: Thank you for your suggestions. We have fixed all these problems.

Point 8: - 202 - screening, not svreening

- 203 - infection, not infestation

Response 8: Thank you for your suggestions. We have fixed all these problems.

Point 9: -- 213-214 - How come WT Arabidopsis also express the transgene BraGSTF2? This result makes me uneasy and leads me to not trust these results. Authors should sequence the qPCR products of the transgenic and WT lines to clarify why WT plants also have an amplicon and if it corresponds to the same gene.

Response 9: Thank you for your comment. The possible reason is that both genes in the cruciferous family, Brassica rapa and Arabidopsis thaliana genes may have the same sequence. However, there is no problem with higher expression in transgenic strains. The F primers of this project partially matched the sequences of Arabidopsis, and the sequences of R primers and Arabidopsis thaliana matched exactly, which should be non-specific amplification of Arabidopsis sequences. But the expression of transgenic strains is higher and the difference is significant, and this is no problem.

Point 10: - 405-406 - were these primes those described in lines 389/390? Clarify in text.

- 441 - genes, not gene.

Response 10: Thank you for your suggestions. We have fixed all these problems.

Point 11: - 196 - introduce bar/scale for cell dimensions

- 387 - Four, instead of 4

Response 11: Thank you for your suggestions. We have fixed all these problems.

Point 12: - 393-394 - provide references for enzyme activities and the other biochemical determinations, providing that you did not make alterations to those protocols. If alterations were done, please describe them.

- 398-404 - format text accordingly

Response 12: Thank you for your suggestions. We have fixed all these problems. The CAT activity and SOD activity, MDA content and SP content was estimated according to the method developed by Gill et al.[46] and Quiroga et al. [47].

Point 13: - References - revise species' scientific names and put them in italic when needed. Also do the same for genes' names.

- 398-404 - format text accordingly

Response 13: Thank you for your suggestions. We have fixed all these problems.

Reviewer 2 Report (Previous Reviewer 1)

Some of the comments have not been addressed

Line 98: No details are provided on the criteria used to classify two BraGSTs as tandem repeats. Authors should provide it in M&M and report these findings in the results section as well. Did the authors find any segmental duplications?

Figure 2: Authors should label the start and end of chromosomes with total length, eg. if ChrA01 is 12 MB, the top should read 0 and the end should read 12 MB.

Figure legend should be more descriptive.

Section 2.7: No empty was used to rule out the non-specific effects i.e. to evaluate if the elements on the 'empty vector' alone (without your inserted gene cassette) will affect the transgenesis result. This is a major flaw in the experimental design.

The addition of new text is in good English, the previous writing still needs editing.

Author Response

Response to Reviewer 2 Comments

Point 1: Line 98: No details are provided on the criteria used to classify two BraGSTs as tandem repeats. Authors should provide it in M&M and report these findings in the results section as well. Did the authors find any segmental duplications?

Response 1: Thank you for your suggestions. Tandem replication of the B. rapaGSTs gene family were analyzed using MCScanX software (make with default parameter values)[42]. We have already added on Line 114-123.

Point 2: Figure 2: Authors should label the start and end of chromosomes with total length, eg. if ChrA01 is 12 MB, the top should read 0 and the end should read 12 MB. Figure legend should be more descriptive.

Response 2: Thank you for your suggestions. We have fixed all these problems. Please see Line 123-128.

Point 3: Section 2.7: No empty was used to rule out the non-specific effects i.e. to evaluate if the elements on the 'empty vector' alone (without your inserted gene cassette) will affect the transgenesis result. This is a major flaw in the experimental design.

Response 3: Thank you for your suggestions. The vector employed in this study is the widely utilized pCAMBIA2300 vector backbone, which exclusively incorporates the Kana resistance gene and the target gene without introducing any additional genes. It is generally presumed that this vector exerts minimal influence on plants.

Round 2

Reviewer 2 Report (Previous Reviewer 1)

Accepted

This manuscript is a resubmission of an earlier submission. The following is a list of the peer review reports and author responses from that submission.

Round 1

Reviewer 1 Report

The manuscript requires extensive editing for sentence structure and grammar as well as font sizes. It is hard for the reader to focus on the findings from the research with the glaring grammatical errors.

Authors should italicize species and genus names throughout the manuscript.

Once a plant name has been intriduced as full form, provide the shorter version thereafter. For eg. after the first mention of Brassica rapa, it should be typed a B. rapa thereafter.

Authors should add more background on types of GSTs and their mode of action in the introduction section to give the reader a better understanding of GSTs role in plant growth and development.

Authors should elaborate how the clustering of GSTs in Figure 1 correlate or differ from their findings using MEME? Did they found same class of GSTs to have same intron-exon structire or same type of motifs? Detailed explanation should be added.

The syntenic relationship of GSTs in Arabidopsis and B. rapa should be explored to understand about the evolution of these genes.

Line 53: What do authors mean by "GSTs involved in gene expression under abiotic stress tolerance in plants"? Please elaborate.

Line 68: Do authors mean 70 in this sentence "A total of BraGSTs gene family were identified and named according to the position of"?

Line 98: No details are provided on the criteris used to classify two BraGSTs as tandem repeats. Authors should provide it in M&M and report these findings in results section as well. Did authors found any segmental duplications?

Figure 2: Authors should label the start and end of chromosomes with total length, for eg. if ChrA01 is 12 MB, top should read 0 and end should read 12 MB.

Figure legend should be more descriptive.

Section 2.5: Did authors just relied on the RNA-seq data to select BraGST2 as potential candidate for future experiments? If yes, the expression should have been validated by qRT-PCR.

Section 2.7: No empty was used to  rule out the non-specific effects i.e. to evaluate if the elements on the 'empty vector' alone (without your inserted gene cassette) will affect the transgenesis result. This is a major flaw in the experimental design.

Line 201: Supplementary figure S3 is not provided in the supplementray file.

Line 306: Provide a link for HMMER and replace 'hmmer' with 'HMMER'

Section 4.4 This section requires extensive scientific editing. The details provided are very generic and lack depth.

Line 342: Which antibiotic?

Line 346: What does authors mean by "4 plants were taken in each pot, and the other part resumed normal growth"?

Figure 6: Need better clarity image.

The manuscript requires extensive editing for sentence structure and grammar as well as font sizes. It is hard for the reader to focus on the findings from the research with the glaring grammatical errors.

Reviewer 2 Report

The manuscript 2473740, entitled "Genome-wide Identification of GSTs Gene Family and 2 Functional Analysis of BraGSTF2 of Winter Rapeseed 3 (Brassica rapa L.) under Cold Stress" contains data that I found relevant and interesting. However, a major english revision should be done to make it readable. I got completely lost during the description of the methodology and the results and, therefore, I do not feel comfortable in evaluating this manuscript as it is.

More notes:

- all species' names should be written in italic;

- you do not need to always write "Brassica rapa" after you wrote it for the first time. Please replace it for "B. rapa", but leave it complete the first you mention this species!

- Regarding the keywords, avoid repeating words that appear on the title;

- revise the text carefully, as there are sentences with erroneous information. Eg.: line 89 - how can 2 genes belong to 4 subfamilies?

- use the same text style uniformly throughout the text;

- what is "Cily copersicum" (lines 46/47)?

As stated above, the quality of english language is very poor. The text is very confusing and difficult to read. A thorough english revision should be done before submitting this manuscript. Pay attention to the grammar, punctuation and written words!